# The “Superoncogene” Myc at the Crossroad between Metabolism and Gene Expression in Glioblastoma Multiforme

**DOI:** 10.3390/ijms24044217

**Published:** 2023-02-20

**Authors:** Chiara Cencioni, Fiorella Scagnoli, Francesco Spallotta, Sergio Nasi, Barbara Illi

**Affiliations:** 1Institute of System Analysis and Informatics—National Research Council (IASI-CNR), 00185 Rome, Italy; 2Axxam SpA, Bresso, 20091 Milan, Italy; 3Department of Biology and Biotechnology Charles Darwin, Sapienza University, 00185 Rome, Italy; 4Istituto Pasteur Italia-Fondazione Cenci Bolognetti, Sapienza University, 00185 Rome, Italy; 5Institute of Molecular Biology and Pathology—National Research Council (IBPM-CNR), 00185 Rome, Italy

**Keywords:** glioblastoma, Myc, metabolic control, gene expression

## Abstract

The concept of the Myc (c-myc, n-myc, l-myc) oncogene as a canonical, DNA-bound transcription factor has consistently changed over the past few years. Indeed, Myc controls gene expression programs at multiple levels: directly binding chromatin and recruiting transcriptional coregulators; modulating the activity of RNA polymerases (RNAPs); and drawing chromatin topology. Therefore, it is evident that Myc deregulation in cancer is a dramatic event. Glioblastoma multiforme (GBM) is the most lethal, still incurable, brain cancer in adults, and it is characterized in most cases by Myc deregulation. Metabolic rewiring typically occurs in cancer cells, and GBM undergoes profound metabolic changes to supply increased energy demand. In nontransformed cells, Myc tightly controls metabolic pathways to maintain cellular homeostasis. Consistently, in Myc-overexpressing cancer cells, including GBM cells, these highly controlled metabolic routes are affected by enhanced Myc activity and show substantial alterations. On the other hand, deregulated cancer metabolism impacts Myc expression and function, placing Myc at the intersection between metabolic pathway activation and gene expression. In this review paper, we summarize the available information on GBM metabolism with a specific focus on the control of the Myc oncogene that, in turn, rules the activation of metabolic signals, ensuring GBM growth.

## 1. Introduction

Despite the great body of knowledge acquired over the past few years, especially at the molecular level, glioblastoma multiforme (GBM), a rare (but the most frequent) brain cancer in adults, still remains an incurable tumor. Survival prognosis is extremely unfavorable—between 15 and 18 months from diagnosis—and depends on both the lack of effective therapies and tumor recurrence [1]. A search of the US clinicaltrials.gov website annotated on 20 October 2022 yielded 1787 clinical studies on GBM, with most of them focused on testing novel or repositioned compounds, such as protein kinase inhibitors, monoclonal antibodies, and small molecules, or strategies, including intraoperative photodynamic therapy, the application of electric fields, and CAR-T immunotherapy. This proves the enormous effort the scientific community is lavishing on methods of novel therapeutic routes for the treatment of this lethal tumor. GBM is characterized by an elevated intratumoral heterogeneity at the single-cell level [2], infiltrative capacity of the surrounding brain tissue linked to neoangiogenesis [3], and the ability to suppress a patient’s immune response by TGF-β upregulation [4]. Three major GBM subtypes may be identified—proneural, classical, and mesenchymal—by converging transcription-based and transcriptomic studies (see below), with the proneural subtype showing the most favorable prognosis but less efficient response to therapy [5]. It is now well-established that, as with other types of cancer, GBM may stem from tumor-initiating cells—the so-called glioblastoma stem cells (GSCs) [6]. Nevertheless, a dedifferentiation process from astrocytes to neural stem cells that, in turn, may transform in GSCs cannot be excluded [7,8]. GSCs resemble the molecular characteristics of the GBM subtype they belong to, suggesting that differential therapeutic targets have to be taken into consideration when approaching a GBM-affected patient. However, within the same tumor, not only can GSCs with distinct molecular signatures be found, but they may evolve and shift from one subtype to another according to the tissue microenvironment (TME) and metabolic cues to which they are exposed [9]. These cells are now considered responsible for recurrency after surgery and resistance to the GBM gold-standard treatment consisting of radiotherapy, followed by adjuvant chemotherapy with temozolomide (TMZ) [1]. Therefore, an in-depth characterization of GSC molecular, biological, and metabolic properties may reasonably represent a promise for the design of more efficient and targeted treatments. Cancer metabolism has to face higher requests for energy supply to sustain enhanced proliferation, migration, and invasion capacity of tumor cells. These latter extract ready-to-use energy from nutrients, mainly glucose, pushing glycolysis, even in the presence of oxygen, known as the so-called Warburg effect [10]. However, tumor cells may also survive glucose starvation, rewiring their metabolism to oxidative phosphorylation (OXPHOS) and lipid oxidation [11,12]. This is also the case of GSCs, which are able to withstand glucose deprivation since they possess residual mitochondrial activity that allows the shift to the OXPHOS and fatty acids oxidation (FAO) pathways. GBM metabolism also relies on the genetic background of a tumor. Indeed, genes that are typically mutated in GBM, such as phosphatase and TENsin homolog deleted on chromosome 10 (PTEN), epidermal growth factor receptor (EGFR), isocitrate dehydrogenase (IDH1), p53, and v-raf murine sarcoma viral oncogene homolog B1 (BRAF), drive downstream metabolic pathways divergent from those found in healthy cells and are strictly dependent on the TME. One of the genes that controls and is controlled as a downstream effector by cell metabolism is the Myc oncogene, which is deregulated in the vast majority of cancers [13]. Myc deregulation may not only depend on genomic alterations in the *myc* locus (insertions, translocation, or amplification [13]) but also on indirect mechanisms, such as post-transcriptional control by microRNAs [14,15] or regulation of myc mRNA and protein stability [16,17,18]. Myc plays a pivotal role in glioblastomagenesis. At the transcriptional level, it is responsible for EGFR upregulation [19] and for the expression or transcriptional repression of a variety of miRs involved in chemoresistance [20,21]. Some Myc tumorigenic effects are exerted through the molecular partners belonging to its protein network [22,23]. More importantly, Myc represents one of the master genes governing the stem cell properties of GSCs [24]. Inhibiting Myc results in a profound resetting of GSC transcriptional nodes, which parallels changes in their biological properties [25].

In addition to Myc’s known response to growth factor stimulation [26], nutrient sensing and hypoxia may also regulate this potent oncogene in healthy cells, both transcriptionally and post-transcriptionally, usually repressing either Myc expression or activity when nutrient and oxygen levels are low [27]. However, this tight, metabolic-dependent regulation of Myc may be bypassed when Myc is overexpressed. Indeed, despite healthy and cancer cells sharing Myc metabolic target genes, upon Myc overexpression, they are deregulated and this leads, at least in part, to cancer cell metabolic rewiring.

Here, based on the available literature to date, we provide a comprehensive view of Myc-dependent control of GBM gene expression profiles and metabolism, highlighting Myc’s role as pivotal regulator of GBM metabolic genes, as well as a target of GBM cell metabolic pathways, in a bidirectional fashion that places Myc at the intersection between transcriptional output and metabolism in glioblastoma.

## 2. Glioblastoma Multiforme

GBM is a grade-IV astrocytoma and, similar to most tumors, is a genetic disease of a single cell. Histopathological analyses identified both the disease stage and the morphological characteristics of the tumor [28]. In the past few years, immunohistochemical analyses flanked, integrated, and supported GBM molecular classification [29], exploiting advanced technologies such as deep convolutional neural networks [30]. GBM is the most malignant brain tumor affecting people of >50 years of age [31], although pediatric GBM may also occur in children. In this latter case, the genetic background is highly different from that found in adults [32]. GBM is defined as rare; however, it represents 14.3% of all central nervous system (CNS) cancers and 49.1% of all malignant CNS tumors [31] in the US and may be classified as “primary” or “secondary”, depending on its “de novo” onset or development from a lower-grade (usually II or III) astrocytoma. Given the similarity of the transcript landscapes among human GBM, radial glial cells, and intermediate progenitors, it is now believed that GBM onset may be due to the reactivation of normal developmental programs [33]. A lack of effective therapies after maximal surgical resection makes this brain neoplasm a never-ending challenge for neuro-oncologists.

### 2.1. GBM Classification

Different GBM subtypes have been identified to date. The term “multiforme” clearly elucidates the morphological heterogeneity of the tumor [34], whose subtypes were first identified as variants by the World Health Organization (WHO) by histopathological characteristics as follows: conventional GBM, giant GBM cell, and gliosarcoma (GS). A number of further histotypes have been identified, as reviewed in Miller CR. 2007 [28].

#### 2.1.1. Genetic Classification

An in-depth characterization in the early 2000s of primary and secondary GBM revealed that these tumors did not differ in their genetic background but, rather, in the frequency of genetic alterations, which was higher in primary GBM with respect to secondary GBM [35]. The major genetic abnormalities found were a loss of heterozygosity (LOH) of chromosome 10q, TP53 mutation, p16^INK4a^ deletion, EGFR amplification, and PTEN mutation. About 10 years ago, the identification of point mutations in the isocitrate dehydrogenase (IDH) 1 (cytosolic) and 2 (mitochondrial) genes [36], provided a further level of classification and, importantly, helped to discriminate between primary (IDH1 wild-type, IDH1wt) and secondary (IDH1 mutated, IDH1mut) GBM. IDH2 mutation at R172 (R172K) residue is found only in 3% of WHO grade-II and -III astrocytomas, oligodendrogliomas, and secondary glioblastomas, whereas IDH1 mutation at R132 (R132H) affects 70% of these cancers [36]. Only 5% of primary GBM shows IDH1 mutation [37]. Highly aggressive IDHwt GBM typically shows EGFR amplification, which has to be considered as a consequence of copy number gains on chromosome 7q. Furthermore, half of EGFR-amplified GBM instances carry a truncated, constitutively active form of the receptor (EGFRvIII), resulting from the deletion of exons 2–7 and encoding a large part of the extracellular domain [38]. Other frequent chromosomal and genetic alterations detected in primary GBM are monosomy of chromosome 10, cyclin-dependent kinase inhibitor (CDKN) 2A and 2B deletion, and telomerase reverse transcriptase (TERT) mutation [39]. A subset of tumors also presents mutations in cyclin-dependent kinases (CDKs) 4 and 6, murine minute doubles (MDMs) 2 and 4, and hepatocyte growth factor receptor (MET) genes [37,40]. Secondary IDHmut GBM also carries α-thalassemia/mental retardation syndrome X-linked (ATRX) mutation [32], together with the mutations mentioned above [35].

#### 2.1.2. Molecular Classification

The coming of high-throughput technologies in the last decades has further detailed the molecular features of GBM subtypes from transcriptional and epigenetic points of view. The term “molecular classification” covers different methodologies used to characterize GBM subtypes, from the analysis of transcriptional and transcriptomic signatures to epigenetic abnormality detection (including genomic methylation profiles) and sensitivity to BUB1 inhibition.

##### Transcriptional/Transcriptomic Classification

One of the first transcriptional classifications of GBM was provided by a microarray analysis of 21 GBM in the context of a larger study also including 45 astrocytic cancers and 19 pilocytic astrocytomas. Among the 6800 genes analyzed, 360 were found to discriminate between GBM and astrocytomas, with some of them (ZYX, SDC1, FLN1, FOXM1, and FOXGB1) previously unidentified as typical GBM genes and related to cell proliferation and migration [41]. A further microarray analysis of 76 grade-III astrocytomas and GBM established distinct, subtype-specific, molecular signatures associated to key stages of neurogenesis, giving emphasis to the presence of stem-like cells as extremely unfavorable from a prognostic point of view [42]. Different probe sets were strongly expressed by three tumor subtypes, and 35 genes for each subtype were identified as robust subtype determinants and used for clustering. From this analysis, three subtypes were identified: proneural (PN), proliferative (Prolif), and mesenchymal (Mes). These reflect both the molecular and the biological features of tumors, with the PN subtype consisting mostly of more differentiated cells and the Prolif and Mes subtypes characterized by genes involved in cell cycle progression control and angiogenesis, respectively. The average expression levels of these 35 genes, termed centroids, were used to evaluate other tumor samples in different databases, strengthening the prognostic value of this classification. Genes depicting Prolif and Mes subtype features were associated to poor prognosis, which was also connected to chromosome 10 LOH, gains in chromosome 7, and activation of the Akt and Notch pathways. Importantly, upon recurrence, a shift from the PN to the Mes subtype is observed with the typical occurrence of YKL-40 gene expression. This gene encodes for an extracellular matrix glycoprotein secreted by different solid tumors and is the most-expressed gene in GBM with respect to healthy brains [43,44,45,46].

The Cancer Genome Atlas network in 2008 provided a comprehensive, multidimensional analysis of copy number, gene expression, and DNA methylation alterations in 206 GBM samples [40]. Together with known genetic aberrations (LOH on chromosome 10, copy number acquisition on chromosome 7, and TP53 mutations), a number of new genomic abnormalities were found. Among them, NF1 inactivation, resulting either from *NF1* gene deletions or point mutations, and mutations in proteins belonging to the PI3K complex were detected. Interestingly, a set of hypermutated tumors was uncovered. These GBM, treated with TMZ or lomustine alone or in combination, presented mutations in genes belonging to the mismatch repair (MMR) pathway. This hypermutator phenotype was also found associated to the promoter methylation of O [6]-methylguanine-DNA methyltransferase (MGMT), which removes alkyl residues from guanine residues in the DNA [47] and predicts GBM response to TMZ [48,49]. Starting from this analysis, Verhaak et al. [5] identified three major GBM subtypes that still represent a reference point for GBM classification at the genomic level: classical, mesenchymal, and proneural. A fourth subtype, called neural, was also described. Each subtype was defined by a set of typical genomic alterations, including single-gene point mutations, and each subtype resembled different neural cell types. Thus, the classical subtype was reminiscent of astrocytes, whereas the proneural subtype was associated with oligodendrocyte signatures. The mesenchymal subtype resembled cultured astroglia, while the neural subtype showed signatures of either astrocytic, oligodendrocytic, or neural differentiation. To simplify, we may say that classical GBM is characterized by EGFR amplification; deletion of the CDKN2A locus; and overexpression of the NES (a neural precursor and stem cell marker), Notch (NOTCH3, JAG1, and LFNG), and Sonic hedgehog (SMO, GAS1, and GLI2) signaling pathways. The mesenchymal subtype presents deletions in *NF1*, as well as overexpressions of YKL-40, MET, TRADD, RELB, and TNFRSF1A belonging to the NK-kB network, probably depending on the infiltrative, inflammatory, and necrotic features of this subtype. The proneural class is invariably identified by alteration in the *PDGFRA* gene and point mutations in *IDH1*. Furthermore, PDGFRA, NKX2-2, and OLIG2—all genes typical of oligodendrocyte development—are highly expressed, making proneural GBM an atypical tumor. The neural subtype is characterized by the expression of neuron-specific genes. However, this subtype is now supposed to derive from contaminated healthy brain cells, as resulted from subsequent analyses [9].

Single-cell RNA sequencing of 596 GBM cells identified after two levels of filtering (for details see Wang Q, Cancer Cell, 2017 [9]) bona fide glioma genes (BFGs). Among them, ~7000 genes matched with the Affymetrix U133A array already used to profile the TCGA cohort [5,40]. This filtered set was used to cluster 369 IDHwt GBM, and the proneural, classical, and mesenchymal subtypes emerged [9]. Further, mesenchymal GBM were transcriptionally characterized by tumor-associated glia and microglia, supporting the hypothesis that GBM subtypes may be shaped by the immune microenvironment. Specularly, *NF1* inactivation was shown to attract microglia, highlighting the bidirectional influence between the tumor and the immune TME.

##### Epigenetic Classification

The term epigenetics refers to heritable—and reversible—changes in the expression of a gene not dependent on alterations in the corresponding DNA sequence. Epigenetic mechanisms are related to the chemical and structural modification of chromatin, including both DNA methylation and histone modifications (acetylation, methylation, ubiquitylation, succinylation, isomerization, phosphorylation, and sumoylation represent some of the 500 histone modifications found to date [50]), as well as histone variants (such as H3.3 [51,52]), RNA-based mechanisms [53], and loops of structurally organized chromatin termed topology-associated domains (TADs), which connect enhancers and gene promoters, coordinating gene expression programs in time and space [54,55]. Histone modifications are catalyzed by chromatin-remodeling enzymes and serve as docking sites for other coregulators, according to the so-called histone code [56], which rules gene activation and silencing. A recent multidimensional omics analysis revealed that genes involved in chromatin organization are mutated in gliomas, in particular in IDHmut gliomas not presenting 1p/19q codeletion [57]. Histone-coding genes have been mostly found mutated in pediatric high-grade gliomas [58,59], whereas the most common epigenetic mark in GBM is a change in the DNA methylation pattern, especially at position 5 of cytosine (5meC) in the context of CpG-rich loci. DNA methylation is invariably associated to gene silencing occurring by recruitment on 5meC residues of methyl-CpG-binding domain (MBD) proteins and, consequently, histone methyltransferases [60]. Both DNA hypo- and hypermethylation may characterize cancer cells. The first is typical of genomic loci that comprise intergenic regions, repetitive DNA sequences, and gene bodies, including oncogenes. The second usually occurs in tumor suppressor genes or the negative regulators of pivotal pathways leading to carcinogenesis. In GBM, for example, DNA hypermethylation is found in genes that negatively control the WNT, Frizzled, and Ras pathways [61,62,63].

*IDH* status profoundly affects glioma genome hypermethylation and is responsible for the glioma CpG island methylator phenotype (G-CIMP) [64]. Although IDH1 R132H mutation affects only 5% of primary GBM [37], IDH1mut proteins represent a striking example of how metabolism and epigenetics are interconnected. IDH is an enzyme of the tricaboxylic acid (TCA) cycle and converts isocitrate into α-ketoglutarate (or D-2-ketoglutarate, D-2-KG) [65]. D-2-KG serves as cofactor for many dioxygenases, including the ten-to-eleven translocation (TET) family of proteins, which demethylate DNA. IDHmut enzymes transform D-2-KG into D-2-hydroxy-ketoglutarate (D-2-HG) [66], which competes with D-2-KG for the active sites of TETs [67], leading to their inhibition and to the acquisition of the G-CIMP hypermethylator phenotype. Furthermore, the IDHmut genotype seems to not allow the proper organization of TADs, with consequent dangerous proximity between strong enhancers and oncogenes, leading to uncontrolled growth [67].

A further level of epigenetic classification has been recently proposed. It takes into account the N6-methyladenosine (m6A) modification landscapes of specific miRNAs in low-grade gliomas (LGGs; in particular, grades II and III) and appears to be more specific in discriminating between low- and high-risk gliomas than IDH status [68]. m6A is the most common RNA modification, affecting both mRNA and noncoding RNAs, including miRNAs, tRNAs, and rRNAs [69,70]. It is catalyzed by writers (such as METTL3/14, WTAP, RBM15/15B, and KIAA1429), recognized by readers (YTHDF1/2/3, IGF2BP1, and HNRNPA2B1), and deleted by erasers (FTO and ALKBH5) [71,72,73,74,75,76]. m6A impacts many aspects of RNA biology (from transcription to processing and from translation to degradation), as well as cell and organ functions. Indeed, it is involved in stress response, metabolism, infectious and metabolic diseases, neural system development, and even cancer [77]. Six m6A-related miRNAs have been identified to have individual prognostic value and to be able to discern between low- and high-risk LGGs. IDHmut and IDHwt LGGs were variably classified as low- or high-risk LGGs using these m6A-related miRNAs as a risk model [68].

##### Classification Based on BUB1B Inhibition Sensitivity

The *BUB1B* gene encodes for the Bub1-like pseudokinase, BubR1, involved in the control of the mitotic checkpoint, timing, and kinetochore–microtubule attachment [78]. *BUB1B* was revealed as essential for GSC expansion [79] and GBM displaying sensitivity to *BUB1B* inhibition (BUB1B^S^) had a worse prognosis when compared to tumors resistant to *BUB1B* inhibition (BUB1B^R^), regardless of molecular subtype. Molecular subnetworks belonging to the BUB1B^S^ state comprise pathways related to cell cycle regulation, microtubule organization, and chromosome segregation. Furthermore, genes in these subnetworks are overexpressed in BUB1B^S^ GBM, explaining the occurrence of mitotic catastrophe in BUB1B^S^ GSCs upon *BUB1B* inhibition [80]. This classification is of particular importance when considering the great heterogeneity within a single GBM and the shift from one molecular subtype to another during tumor relapse.

### 2.2. Glioblastoma Stem Cells (GSCs)

Cancer stem cells (CSCs)—also termed tumor-initiating cells (TICs)—were first described in hematologic malignancies [81] but were found immediately after also in solid cancers [82], including GBM [83]. These cells represent a minor population within the tumor mass. Nevertheless, GSCs are responsible, in the vast majority of cases, for tumor recurrence and resistance to therapies. GSCs recapitulate most of the molecular characteristics of pluripotent stem cells, including the activation of stem factors such as Oct4, KLF4, Nanog, and SOX2, which constitute a self-fueled transcriptional circuit [84,85,86], and the Wnt/β-catenin, Sonic Hedgehog (Shh), and Notch pathways [87,88,89], granting self-renewal. They also possess the capacity to undergo asymmetric cell division, giving rise to both proliferating, self-renewing stem cells and to differentiated cells with low tumorigenic potential, which enrich the bulk of tumors. Nevertheless, this process is not so efficient due to the small amount of CSCs harboring this ability [90,91,92]. Given the infiltrative nature of GBM, surgical eradication of GSCs is always incomplete; further, a number of efficient mechanisms, including DNA repair systems, as well as the Notch, NF-κB, EZH2, and PARP pathways [89,93,94,95], are engaged by these cells when exposed to radiation and chemotherapeutic agents, escaping apoptosis [96]. By definition, GSCs present self-renewal and multipotency properties owned by neural stem cells [97] and recapitulated tumor heterogeneity when implanted in immunocompromised mice [98]. They may derive from neural stem or progenitor cells or from a dedifferentiation process occurring in glial cells [99], with this latter mechanism still an object of debate. Further, they resemble the tumor subtype of origin in terms of molecular features, harbored genetic lesions, and biological behavior [5,100]. How many subtypes of GSCs exist is still controversial. Three GBM subtypes have been described (see Transcriptional/Transcriptomic Classification), but some papers have reported that mainly the mesenchymal and proneural subtypes of GSCs may be detected within tumors [101,102,103]. Furthermore, although the neural subtype has been referred as a simple contamination from healthy surrounding brain tissue [9], a very recent work demonstrated that this specific GSC subtype may be present at the outer invading border of the tumor as a result of the activation of a neuronal developmental program elicited by signals originated from neighboring neurons [104].

Furthermore, GSCs have been considered responsible for GBM intratumoral genomic heterogeneity [105,106]. Indeed, genomic heterogeneity and evolution have been observed in both GBM patients and mouse xenografts [5,37,107,108]. Therefore, it is conceivable that these phenomena could depend, at least in part, on genomic variations in GBM cells of origin, leading to the acquisition of fitness advantages. Nevertheless, it was demonstrated that GSC heterogeneity acquired through serial transplantation in mice of homogenous populations did not depend on heritable epigenetic or genetic changes but rather on growth dynamics and cell fate decisions [109]. When looking at TMZ resistance, however, epigenetic variations have been found to distinguish between resistant vs. nonresistant clones. In this case, changes in histone H3.3 level have been found, corresponding to downregulation of the encoding gene [109,110].

#### 2.2.1. GSCs and the Tumor Microenvironment

The TME is particularly important for tumors, as it provides nutrients. GBM evolution is strictly dependent on the interaction with the TME, especially TME immune components. It has been revealed that distinct GSC subtypes evoke transcriptional signatures typical of different immune cells. Indeed, by single-cell sequencing, the mesenchymal subtype was found associated with transcript outputs belonging to M1 and M2 macrophages and neutrophils, whilst natural killer (NK) transcripts were almost absent. The proneural subtype was depleted of memory CD4+ T lymphocytes, whereas the classical subtype (according to molecular classification in the TCGA network, Nature 2008 and in Verhaak, R.G., Cell, 2009 [5,40]) was enriched in dendritic gene signatures [9]. Importantly, the immune TME is subjected to GSC-subtype-specific variations upon recurrence. Thus, recurrent mesenchymal GBM showed decreased levels of associated monocytes but increased M2 macrophages. Classical and proneural GBM were characterized by a global decrease in immune cells defined by their specific transcriptomic landscapes, but when shifting to the mesenchymal subtype, an increase in the TME immune fraction was found. Notably, when hypermutation was sustained by TMZ treatment, enrichment in CD8+ T lymphocytes was detected [9].

Interaction with TME nonimmune components may activate surprisingly normal transcriptional programs in GSCs. This phenomenon is particularly evident at tumor relapse. Indeed, in IDHwt GBM where a stem-like state transcriptional signature is found, GSCs located at the outer invasive border present a typical neuronal gene expression pattern, which lacks GSCs at the core of the tumor. In fact, the neuronal marker NeuN, the stem factors SOX2 and SNAP25, a genes expressed by GSCs upon recurrence are present in the infiltrative region, whereas SNAP25 is absent and few neurons are present in the tumor core. This suggests that the neuronal-like phenotype, characterizing GSCs at the tumor infiltrative border, is the result of signals that hit GSCs from the surrounding neurons in the invaded brain. Conversely, IDHmut tumors—characterized by a differentiated-like state signature—differentially express genes belonging to cell cycle and mitosis categories [104].

These studies clearly show that GBM—and the GSCs within—and the immune microenvironment co-evolve. Indeed, by sequencing 61,062 single cells from eight IDHwt GBM cases, a natural evolution signature (NES) has been identified. This signature is a module of 12 genes and is associated to the activation of brain developmental genes, such as MYBL2 and FOSL2. The higher the presence of the NES in a tumor, the older the lesion. Furthermore, the NES is promoted by hypoxia and associated with the infiltration of bone-marrow-derived macrophages in the pseudopalisading cell region (PC) and microvascular proliferating region (MP). The infiltration of bone-marrow-derived macrophages matches with the expression of a specific gene expression signature. Among the genes differentially expressed, *ANXA1* has been revealed to have a pivotal role in recruiting monocytes at tumor sites and inducing their differentiation into M2-like macrophages, which are typically immunosuppressive [111,112]. This resulted in a low proliferative rate and low INFγ production of CD8+ lymphocytes co-cultured in vitro with ANXA1+ tumor M2 macrophages, as well as in increased tumor mass in orthotopic mouse xenografts [113].

#### 2.2.2. GSC Metabolism

GSC metabolism is highly divergent with respect to bulk tumors (see Section 3 below). Indeed, it has been reported that they are characterized by lower glycolysis and maximal respiratory capacity. Moreover, GSCs present globally lower levels of metabolites (such as GTP and ATP), nonessential amino acids, fructose 6-phosphate, and fructose 1,6-bisphosphate. However, metabolites belonging to the pentose phosphate and galactose metabolism, glutaminolysis, and the citric acid cycle are present at high levels in GSCs [114]. Notably, the metabolic profile reflects GSC subtype identity. Low mobile lipids and high glutamine consumption define the GSC restricted-stem (GSr) phenotype [115,116], whereas the GSC full-stem (GSf) phenotype presents the opposite [101]. These typical phenotypes correspond to the proneural and the mesenchymal subtypes, respectively, according to the molecular classification of Verhaak et al., Cell, 2009 [5]. Overall, two metabolic classes of GSCs may be identified: one dependent on glucose and aerobic glycolysis to support proliferation and the second sustaining OXPHOS through glutamine and activating glycolysis only when glutamine is limited [117]. Importantly, GSC metabolism reflects resistance to therapies and may shift upon treatment. Active lipid and glutamine metabolism, low glucose consumption, and OXPHOS have been found correlated to resistance to radio- and chemotherapies [118,119,120]. When irradiated, GSCs still showed their quiescent metabolic state, with little change in adenylate energy charge, glutamine/glutamate ratio, and amino acid profile [114].

## 3. GBM Metabolism

GBM presents an altered metabolic state at multiple levels. Glucose, lipid, and fatty acid metabolic pathways exhibit relevant alterations in GBM, which depend on different cues. Even cell location in a tumor mass may dictate metabolic rewiring [121], thus making GBM a heterogenous tumor also from a metabolic point of view. Moreover, GBM metabolism has to be considered as the result of a complex networking of different cellular components within the CNS [122,123]. This is particularly important for IDHmut GBM, which, deriving from a lower-grade neoplasia, is considered a metabolic disease.

Activation of distinct metabolic routes also depend on the GBM genomic and epigenomic landscape, as specific mutations are associated with typical metabolic pathways, whilst perturbations in chromatin-remodeling enzyme activity may impact metabolic changes and vice versa. These specific topics have been extensively reviewed elsewhere [124,125,126].

### 3.1. Glucose Metabolism

The brain almost entirely relies on glucose uptake as the main source of energy. However, glucose influx into the brain is hampered by the presence of the blood–brain barrier. To counteract limited glucose availability, neurons express GLUT3 (*SLC2A3*) glucose transporter, which has a five-fold higher affinity for glucose. Tumors, including GBM, mainly transport glucose through GLUT1. However, recently it was demonstrated that GSCs preferentially express GLUT3, which supports GSC tumorigenic properties [127]. As with other cancer types, GBM uses aerobic glycolysis to fulfill energy demand and is characterized by a high glucose uptake, which undergoes cytosolic fermentation resulting in an overproduction of lactate released in the intercellular space [128], stimulating angiogenesis and impairing immune surveillance [129,130]. High glycolytic rate—occurring especially in the tumor core—has been associated with dismal prognosis and poor survival of GBM patients [11,131]. However, glucose consumption through glycolysis is not a “dogma” and, depending on a multiplicity of factors (interaction with the TME, cell cycle phase, and oxygen levels), GBM metabolism may switch to OXPHOS, typically at the tumor outer layer [10]. Interactions with the immune components of a tumor mass play a pivotal role in GBM metabolic adaptation [132], pushing GBM through glycolysis. Among the molecular pathways that contribute to enhance glycolysis in GBM, there is the upregulation of human ribonucleoprotein (hRNP) A1, which promotes the alternative splicing of Myc-associated factor X (Max), generating Delta-Max, which is typical of EGFRvIII-mutated GBM. Delta-Max enhances Myc-dependent transformation and supports glycolytic gene expression. Indeed, expression of Delta-Max counterbalances EGFRvIII loss. It also induces EGFRvIII expression, originating a transcriptional positive feedback loop whose intermediate is hRNPA1, and fuels GBM cell proliferation [133]. Overexpression of isoform 2 of hexokinase (HK2), the first enzyme of the glycolytic pathway, characterizes GBM-carrying PTEN mutations, as well. HK2 is not an isoform typically expressed by the brain, which is characterized by HK1. Inhibition of HK2 restrains GBM proliferation, reversing the Warburg effect. Sensitization to radiation and TMZ also occur upon HK2 depletion [134]. Furthermore, the dissociation of HK2 from mitochondria in the presence of high glucose leads to the activation of the NK-kB pathway through IkBα binding, phosphorylation, and degradation by μ–calpain. Activated NF-kB induces the expression of PD-L1, connecting GBM metabolism with immune evasion [135]. Blockage of cytochrome oxidase 2 synthesis, which inhibits glycolysis, and upregulation of glucose transporters (SLC2A1 and A4) [136,137] have been found in p53-mutated GBM. The TCA cycle and downstream glycolysis also present large perturbations in GBM. In IDHwt tumors, IDH is overexpressed compared to healthy brain tissue, and this leads to the production of high levels of D-2-KG using NADP+ or NAD+ as cofactors. D-2-KG, in turn, serves as a cofactor for the ten-to-eleven translocation (TET) family of demethylating enzymes [67]. As a consequence, in IDHmut secondary GBM, a novel metabolite has been found called D-2-HG, whose detection requires specific methods that are able to distinguish between the “D” and the “L” isomers, which differ by an asymmetric carbon atom [138]. Only D-2-HG may compete with D-2-KG for TET binding, inhibiting its activity and leading to the “hypermethylator phenotype” associated with a better prognosis [64]. These findings are a clear demonstration of how metabolic alterations directly reflect the epigenome.

### 3.2. Glutamine Addiction

When glucose is limited, GBM cells utilize glutamine as a primary energy source. Glutamine is essential for the synthesis of final metabolic products (amino acids, proteins, purine/pyrimidines, and hexosamine), as well as cofactors, such as NAD. It is also required to refuel the TCA cycle of intermediates subtracted for the synthesis of compounds under nutrient deprivation conditions, activating catabolism through a series of reactions not requiring Acetyl-CoA. Glutamine also activates the mechanistic target of rapamycin (mTOR) and controls cellular pH [139,140]. Glutamine levels are very high in GBM [141], and this is due to the overexpression of glutamine synthetase (GS), which produces glutamine from glutamate and ammonia [142]. Conversely, glutaminase (GLS) hydrolyzes glutamine into glutamate and ammonia. This is the first step of glutaminolysis, leading to the production of D-2-KG from glutamate via glutamate dehydrogenase (GDH). It has to be mentioned that pyruvate carboxylation may be also used to resupply TCA cycle intermediates, specifically citrate resulting from the condensation of Acetyl-CoA and oxaloacetate [143]. However, this reaction is downregulated in most cancers [144], and glutaminolysis remains the major source of D-2-KG to push the TCA cycle. Glutamine may also be transformed via glutamate into proline, which has gained attention in recent years as a modulator of cancer biology [145]. Proline levels were found elevated in GBM because of its reduced catabolism due to decreased expression of proline hydroxylase (POX) [146]. In addition to D-2-KG, glutathione (GSH) and lactate are the end-products of glutamate. GSH is of major importance because, as a potent antioxidant, it is one of the main responsible factors for GBM resistance to therapies. Indeed, GSH depletion has been explored as a putative strategy for GBM treatment [147,148]. Glutamine-derived lactate essentially recapitulates the Warburg effect through the oxidative decarboxylation of malate to pyruvate CO_2_ and NADPH, which enter glycolysis via glyceraldehyde triphosphate dehydrogenase (GAPDH) [149].

### 3.3. Lipid Metabolism

The brain is the second organ, after adipose tissue, to rely on lipids to build its structure, extract energy, and maintain cell growth [150]. The lipidomic profiles of GBM cells and tissue derived from mouse xenografts were different [151], but human gliomas present in large proportions sphingomyelin, phosphatidylinositol, and lysophosphoglycerides, as well as increases in phospholipids, cholesterol esters, arachidonic acid, and oleic and linoleic acids, when compared to a healthy brain cortex [151,152]. Lipid biosynthesis requires fatty acids that are, indeed, elevated in malignant brain tissue [153]. Acetyl-CoA synthesis is a fundamental prerequisite to produce fatty acids, and most GBM tumors present an upregulation of Acetyl-CoA synthetase 2 (ACCS2), which converts acetate into Acetyl-CoA [154,155]. Activation of Acetyl-CoA carboxylase (ACACA) via the EGFR/PI3K/Akt pathway also enhances lipogenesis in GBM [156]. FAO is a pivotal process to foster GBM growth, as inhibiting fatty acid oxidation slows the proliferation rate of glioma cells both in vitro and in vivo [12,157]. Importantly, lipid metabolism is linked to a newly discovered form of programmed cell death, ferroptosis, which depends on enhanced iron-dependent lipid peroxidation and ROS production, a route that is being explored to treat GBM [158,159].

## 4. Myc

Almost 40 years ago, the transforming MC29 avian virus sequence causing myelocytomatosis, later termed *myc*, was identified [160]. The finding of the human homolog of the viral avian *myc* gene [161] served as the basis for the work of entire laboratories worldwide, which spent the following decades elucidating Myc biology, profoundly marking cancer research [13]. The finding that *myc* gene deregulation in tumors did not occur through mutations in the coding region—as with other oncogenes, such as *ras*—was surprising. Indeed, *myc* deregulation in cancer occurs through three different mechanisms: gene amplification [162], chromosomal translocation [163], and insertional mutagenesis [164]. Three *myc* paralogs have been identified, including *c-myc* (hereafter termed *myc*), *n-myc* [165], and *l-myc* [166], which are invariably deregulated in a wide variety of cancers in a tissue-specific manner, including GBM.

### 4.1. Brief Overview on Myc Structure, Post-Transcriptional Regulation, and Control of Gene Expression

Myc in all its three isoforms is a basic helix-loop-helix-leucine zipper (bHLH-LZ) transcription factor, an object of extensive review elsewhere [13,167]. It binds almost 15% of genomic loci and is deregulated in 70% of cancers. Here, we provide a snapshot of its structure and regulation and how it works in the control of gene expression, both directly and indirectly.

#### 4.1.1. Myc Structure

Myc is a 439 amino acid protein. The encoding gene is located on the long arm of chromosome 8 (8q24) and contains three exons, which encode a translational product of 64KD. The two highly conserved Myc polypeptides consist of a N-terminal transactivating domain (TAD) and a C-terminal DNA-binding region. The TAD is constituted by three conserved Myc boxes (MB0, MBI, and MBII) [168,169]. MBI and MBII are required for transcriptional activation. In particular, MBI recruits P-TEFb, a cyclin CDK-complex promoting transcriptional elongation by RNA polymerase II (RNAPII) phosphorylation [170]. MBII is essential for Myc targeting of transcriptional activation and repression. MBII also plays a role in promoting cellular transformation and tumorigenesis; it also controls Myc turnover. Myc structure also presents a middle proline, glutamic acid, serine, and threonine (PEST)-rich region adjacent to the MBIII and MBIV conserved boxes and two nuclear localization sequences. The MBIII box, by recruiting histone deacetylase 3 (HDAC3), is engaged in transcriptional repression, whereas Myc pro-apoptotic function seems to rely on MBIV. The 100-amino-acid carboxyterminal region is a bHLH-LZ domain required for heterodimerization with Max, another small bHLH-LZ protein [171]. This interaction forms a stable Myc/Max heterodimer that directly binds specific DNA sequences named “enhancer boxes” (E-boxes) to stimulate transcription. Indeed, Myc homodimers are highly unstable [172].

#### 4.1.2. Myc Protein Network

Myc-driven proliferation, transformation, and apoptosis processes require Myc/Max heterodimerization. Max is a stable, constitutively expressed transcription factor (TF) with a short half-life. This suggests that Max activity is highly dependent on the amount of Max-associated TFs. Max homodimers, by competition for E-box binding, may impair Myc biological activity. Other Max interactors are Mad1–4 (Mad1, Mxi1, Mad3, and Mad4), Mnt, and Mga, also provided with a bHLH-LZ domain. These proteins share some Myc properties: (a) scarce homodimerization and DNA-binding ability; (b) robust heterodimerization with Max and consequent E-box binding; and (c) interference with Myc/Max heterodimer activity. The family of Mad1, Mxi1, and their related members of transcriptional repressors competes for Myc/Max binding sites. The competition of Myc and Mad for Max defines a cell’s choice between proliferation and transformation and differentiation and quiescence [173], as increased expressions of Mads determine cellular differentiation and growth arrest. Indeed, Mad1–4 proteins possess a repression motif interacting with Sin3a and Sin3b corepressors, which in turn engage HDACs and other chromatin remodelers. In conclusion, the opposite functions of Myc and Mad are exerted at multiple levels: (i) competition with Max to form heterodimers; (ii) competition for E-box binding; and (iii) transcriptional activation and repression of target genes [174]. Interestingly, among Max partners, Mnt is a unique Myc antagonist. Its expression overlaps with the expression of Myc, but it is constitutively expressed [175,176]. Furthermore, Myc can also repress the transcription of specific subsets of genes competing with p300 for binding to the transcriptional activator Miz1 [177].

#### 4.1.3. Myc Post-Translational Regulation

Myc undergoes extensive post-translational regulation, which is required to both switch on and off its activity. MBI, -II, and -III in its N-terminal TAD domain are involved in protein stabilization and transcriptional activity [178]. A series of phosphorylation and dephosphorylation events occurs at MBI, where serine 62 (S62) is phosphorylated by RAS/MEK/ERK/CDK2 [179], which precedes and induces the phosphorylation of threonine 58 (T58) by GSK3β [18]. Recently, S67 has been found to be the target of Aurora B kinase. This event blocks T58 phosphorylation by GSK3β, leading to enhanced protein stability [180]. Indeed, following T58 phosphorylation, S62 is dephosphorylated by Pin1/PP2A, and this promotes Myc degradation and turnover through the SCF^FBXW7^/Ub-proteasome pathway [18]. In addition to the SCF^FBXW7^/Ub-proteasome pathway, Myc may also be degraded by the SCF-SKP2 ubiquitin-ligase complex [181]. SKP2–Myc interaction is intriguing, as SKP2 and its related components of the SCF complex are first recruited at Myc target promoters and, after Myc binding and ubiquitination, transcriptional activation of Myc occurs before its degradation through the proteasome, finely tuning Myc transcription in a very narrow window [182]. Notably, a third ubiquitin ligase, named HECTH9, was found to promote Myc transcription rather than its degradation by polyubiquitination and polymerization of lysine (K) 63, enhancing K acetyltransferase (KAT) recruitment [183].

Myc ubiquitination is interconnected with its acetylation by KATs, such as p300, mammalian GCN5 (mGCN5), and Tip60 [184,185]. Myc holds many Ks that may be acetylated, and at least one of them, K323, is located in the NLS [186]. However, acetylation does not alter Myc localization or Max binding; rather, p300-dependent acetylation of Myc has been revealed to promote a more efficient recruitment of Myc-interacting zinc finger protein (Miz1) to Myc/Max heterodimers.

Most recently, it has been demonstrated that Myc may also be methylated on arginine (R) residues by the protein arginine methyltransferase (PRMT) family of enzymes. Class-I PRMT monomethylates or asymmetrically dimethylates R residues on target histone and nonhistone proteins. Class-II PRMT monomethylates and symmetrically dimethylates R residues on targets [187]. In particular, both asymmetrical and symmetrical Myc dimethylation occurs, having opposite roles on Myc properties. In M2 macrophages, Myc asymmetrical dimethylation by PRMT1 leads to the efficient recruitment of p300, enhancing transcription [188]. In GSCs, both asymmetrical and symmetrical Myc dimethylation has been detected. These modifications differently impact Myc stability in GSCs and rule differentiation and stemness, respectively [189]. Figure 1 summarizes Myc structure, post-translational modifications, and interactions with protein partners.

#### 4.1.4. Control of Gene Expression by Myc

The concept of Myc as a “classical” transcription factor that binds to DNA and promotes transcription through the recruitment of KATs—which untangle compacted chromatin, acetylating histones—and components of the basal transcriptional machinery has consistently changed over the last decade. Indeed, Myc unusually controls all three RNA polymerases (RNAPs), thus controlling the synthesis not only of encoding and noncoding RNAs, but also of tRNAs and rRNA, which are pivotal to protein biosynthesis upregulation in tumor cells. Of particular note are the recent findings of a tight control of RNAPII by Myc. In fact, it has been demonstrated that Myc regulates RNAPII pause, release, and elongation [170,190] and also maintains the fidelity of splicing in cancer cells [191]. Furthermore, Myc was postulated as a global transcriptional amplifier of already active gene expression programs [192], especially depending on the amount of molecules within a given cell (Figure 2). Indeed, in cancer cells, high Myc levels drive transcriptional amplification, potentially invading chromatin regulatory elements (promoters, enhancers, and superenhancers) with both high and low affinity for Myc binding [193]. The degree of Myc occupancy determines the expression level of each active gene: high-affinity promoters are already saturated by Myc in proliferating cells, and a further increase in Myc molecules only enhances its occupancy at low-affinity binding sites [194].

This behavior of Myc is consistent with the evidence that open chromatin at active promoters is important for Myc binding [195] and that enhancer loops in the proximity of core promoters at active genes may facilitate Myc recruitment to close enhancer elements once binding sites in core promoters are saturated. It has to be noted that Myc-bound and Myc-regulated genes do not always overlap. An explanation of this phenomenon resides in subsets of microRNAs regulated by Myc and clustered according to the specific role their targets play in Myc-controlled cellular processes [196].

Myc also regulates mRNA cap methylation, an essential step of mRNA translation. In fact, Myc recruits TFIIH kinase, which phosphorylates RNAPII that, in turn, recruits and activates cap RNA methyltransferase (RNMT) [197]. Finally, although generally considered a transcriptional activator, Myc was also found to repress transcription in complex with Miz1 in a ratio near to 1 at Myc-repressed promoters [194].

#### 4.1.5. Myc-Dependent Regulation of Apoptosis

In healthy cells, a high concentration of growth factors is responsible for cell proliferation in response to sustained Myc levels; when growth factors are limited, Myc primes cells to undergo apoptosis [198,199]. Conversely, deregulated Myc expression is responsible for resistance to apoptosis in transformed cells [200,201,202]. Although the pro-apoptotic function of Myc is not precisely defined at the mechanistic level, multiple pathways seem to be involved, and it is not clear whether Myc constitutively modulates downstream effectors or whether second stimuli are needed for the full activation of each apoptotic pathway. Myc was demonstrated to activate p53 [203], partially depending on p19^ARF^ induction and activity and, consequently, MDM2 and p21 activation [200]. However, Myc was also found to repress p21, overcoming the G1-S cell cycle checkpoint by both directly binding its promoter in complex with Miz1 [204] and competing with Sp1/Sp3 transcription factors [205]. Myc may also alter the balance between anti-apoptotic and pro-apoptotic pathways in favor of the latter, when required. For example, Myc acts as a suppressor of Bcl2 and Bcl-XL [206,207], promoting apoptosis through Bax [208,209] and influencing cytochrome c release from mitochondria [210] with the consequent activation of caspases [211].

Further, under certain conditions, Myc may promote cellular senescence, a phenomenon blunting cancer development at the premalignant stage [212,213].

### 4.2. Myc and the Control of Gene Expression in Glioblastoma

As in other cancer types, Myc is usually overexpressed in GBM. Indeed, genes pivotal to glioblastoma progression and resistance to therapies are aberrantly regulated by Myc in GBM. Myc binds to the promoter of EGFR, which, in turn, regulates Myc in a positive feedback loop. An overexpressed EGFR/myc axis results in the activation of the TGF-β, Notch, and Hippo pathways through the downregulation of mir524 [19]. Furthermore, EGFRvIII induces glioma angiogenesis by Myc-dependent transcriptional activation of angiopoietin-4 (ANG4) (see Crosstalk between Myc-Dependent Angiogenesis and Metabolism in GBM below) [214]. At the epigenetic level, lysine demethylase 4 (KDM4) binds to the Myc promoter, inducing its expression [215]. The same finding was obtained for histone methyltransferase G9a, which was revealed to induce Myc gene expression and the proliferation and invasiveness of GBM cells [216].

Far-upstream element (FUSE)-binding protein 1 (FBP1), RANBP10, and Trip13 all regulate Myc in GBM. FBP1 directly binds to the Myc promoter [23], whereas RANBP10 and Trip13 both increase Myc stability by downregulating FBXW7 [22,217]. SWD3, a WD repeat domain 5 (WDR5) protein, also regulates MYC activity. The Myc/SWD3 complex is recruited to the PRMT4 (CARM1) promoter, and PRMT4 expression enhances glioblastoma proliferation [218]. Interestingly, in GBM Myc is regulated by the lncRNA HOXC13, which, acting as a sponge for mir122-5p, activates SATB1 and Myc. Myc, in turn, is able to bind to the HOXC13 promoter, generating a positive feedback loop sustaining GBM invasive properties [219].

A number of microRNAs are controlled by Myc in GBM. Interestingly, most of them are related to the acquisition of chemoresistance. Indeed, Myc negatively controls mir29c expression, an oncosuppressor whose expression is inversely correlated to tumor cell proliferation and invasion [220]. Myc-dependent mir29c downregulation is paralleled to the TMZ resistance of GBM cells [21]. Myc also enhances mir20a expression, pushing cell proliferation and chemoresistance [20]. More importantly, Myc directly binds to the MGMT promoter [221]. Myc activity also regulates GBM epithelial–mesenchymal transition through wnt/β-catenin signaling [222].

### 4.3. Myc and the Maintenance of GSCs

Myc exerts one of its major roles in glioblastomagenesis by controlling the proliferation and self-renewal of GSCs, being located at multiple GSC genomic loci involved in metabolism, protein synthesis, and the cell cycle. This is a feature of Myc not restricted to CSCs, as in embryonic stem (ES) cells, Myc exerts the same role, while pluripotency is governed by core transcription factors including OCT4, Nanog, and SOX2 [223]. Indeed, it has been found that, in ES cells, gene modules related to the core of pluripotency and to the Myc network are independent. The Myc module is centered on Myc–NuA4 interaction. NuA4 is a HAT complex comprising Tip60 and Ep400. In ES cells, the latter have been found associated to Myc/Max heterodimers, as well as with Dmap and Trrap, which keep chromatin in the open state, promoting massive histone acetylation. The Myc cluster includes transcription factors, such as E2F1 and 4, Rex1, Zfx, and N-Myc, which invariably are associated with hyperacetylated chromatin. Interestingly, this Myc module is found also in cancer stem cells, where it does not activate the core module [224], confirming the independency between self-renewal and proliferation potential and pluripotency. Consistently, Myc gene depletion blocks GSC proliferation and capacity to form neurospheres, whilst bulk tumor cells are less dependent on Myc expression [24]. Myc interactions with E2F3 and the chromatin regulator helicase, lymphoid-specific (HELLS) support the expression of genes critical to GSC maintenance [225].

Inhibiting Myc protein interactions and DNA-binding abilities results in a resetting of specific gene regulatory nodes in GSCs, with the highest impact on gene modules ruling stemness and neural development. This is consistently mirrored at the biological level with changes in prominent GSC features, such as enhanced proliferation and differentiation capacity [25].

Myc is regulated in GSCs in a number of ways. Very recently it was demonstrated that, in GSCs, Myc mRNA is methylated on adenosine (N6methyladenosine; m6A) and stabilized by the m6ARNA reader YTHDF2, which is itself indispensable for the expression of Myc target genes and GSC maintenance [226]. Myc-sustained expression is also regulated by Piwi1, which belongs to the family of small, RNA-binding argonaute proteins. Indeed, in the presence of Piwi1, the expression of FBXW7 is low, and this leads to a decrease in Myc degradation [227]. Consistently, deubiquitinases, such as USP13, maintain GSC-impairing proteasome-mediated Myc degradation [228]. In addition, cycline-dependent kinase 8 (CDK8) has a role in maintaining high Myc levels in GSCs, although the mechanisms still have to be elucidated [229].

The immune TME also plays an important role in regulating Myc activity. Indeed, it has been recently found that β2-microglobulin activates Myc through the PI3K/AKT/mTOR pathway, inducing the secretion of TGF-β. The latter, in turn, activates SMAD and PI3K/AKT signals in M2 macrophages, promoting the consolidation of a immunosuppressive tumor microenvironment [230].

### 4.4. Myc-Targeted Therapies in GBM

Being deregulated in >70% of tumor types, Myc targeting for anticancer purposes has been, and still represents, a challenge for both biologists and clinicians. Inhibiting Myc activity has to be balanced with the pivotal functions Myc exerts in healthy cells. Therefore, the complete abolishment of Myc activity through gene-silencing techniques would have detrimental side effects in the whole organism. Nevertheless, impacting Myc ability to form protein complexes with molecular partners and targeting Myc transcriptional coregulators have been explored as evaluable strategies to be translated to clinics (reviewed in Chen H, 2018 [231]; Llombart V, 2022 [232]). Clinical trials specifically related to Myc-overexpressing brain tumors are listed in Table 1 and show that limited actions are currently available to hit Myc in GBM. One of the most promising molecules affecting glioblastoma cell viability is TG02, a CDK inhibitor that is supposed to act through the CDK9-inhibition-dependent downregulation of oncoproteins, including Myc. TG02-treated cells undergo mainly apoptosis. However, blocking caspases does not result either in apoptosis impairment or in Myc expression recovery. Interestingly, TG02 activity is independent from MGMT expression, and repetitive exposure to TG02 does not induce resistance in GBM cells [233]. Inhibition of the bromodomain-containing chromatin modifier BRD4 is another exploited strategy to treat solid and hematologic malignancies characterized by high levels of Myc. This therapeutic route is based on the reciprocal control between Myc and BRD4. BRD4 destabilizes Myc by direct interaction [234]; conversely, Myc stimulates its own transcription via BRD4 recruitment and histone acetylation within its promoter [235]. JQ1 was one of the first bromodomain and extraterminal domain (BET) inhibitors used for clinical purposes and has been found effective in halting tumor growth in a variety of cancers by limiting *myc* expression [236,237]. However, its pharmacokinetic properties have precluded its use in clinical trials. Other BET inhibitors have been tested and have reached phase-I and -II clinical trials. INCB057643 is a novel BET inhibitor that is orally available and was already tested in phase-I clinical trials in patients with myelofibrosis and advanced malignancies [238], including GBM (Table 1; https://clinicaltrials.gov/ct2/results?cond=Glioblastoma&term=myc&cntry=&state=&city=&dist=&Search=Search; accessed on 12 January 2023). Recently, its application was extended to preclinical models of pancreatic cancer, where it was shown to modulate the immune TME [239].

## 5. Myc and Metabolism in Cancer

Independently of Myc expression levels, healthy and cancer cells share Myc target genes involved in many metabolic pathways, such as glycolysis, glutaminolysis, and lipid and nucleotide synthesis. In healthy cells, Myc and related metabolic pathways are tightly regulated, especially upon nutrient deprivation. In neoplastic cells, derangements at the genetic and epigenetic levels and loss of checkpoints (e.g., p53 and PTEN) that usually restrain dangerous events depending on Myc overexpression or activation allow the occurrence of metabolic Myc activities that promote cell growth.

In nontransformed mammalian cells, Myc metabolic activity is regulated by multiple factors. In optimal nutrient conditions, Myc drives metabolic pathways in order to provide cellular bricks and energy required to replicate DNA, undergo cell division, and increase cell mass. In addition growth factors, which are known to stimulate Myc [240], nutrient sensing controls Myc activity through mTOR, which modulates its translation and its stability [241,242]. Active FOXO3A inhibits Myc in a number of ways: it transactivates Myc antagonist max interactor-1 (MXI-1), which displaces Myc from Max, inhibiting Myc/Max transcriptional activation, and it also counteracts Myc-dependent transactivation of mitochondrial genes, affecting mitochondrial biogenesis [243,244]. Hypoxia and glucose deprivation are two other factors that negatively influence Myc expression through degradation or antagonist expression [245,246].

Cancer metabolism is one of the best proofs of Myc as a transcriptional amplifier. Indeed, metabolic genes are expressed at steady-state levels in almost all cells. To fulfill tumor cells’ enhanced energy demand, overexpressed Myc aberrantly amplifies genes devoted to glycolysis, glutaminolysis, and polyamine synthesis [247,248,249]. Consistently, most of the metabolic genes controlled by Myc are provided with E-boxes [250,251], and genes holding low-affinity E-boxes may be regulated by Myc as well when it is overexpressed [252]. Myc not only controls the expressions of enzymes devoted to glucose and glutamine metabolism—such as the glucose transporter GLUT1 and the glutamine transporter SLC1A5 [248,253]—but it also regulates the preferential splicing of variants [254]. Upregulation of the nutrient-sensing transcription factors MondoA and ChREBP, which control different aspects of cellular metabolism and accumulate in the nucleus depending on metabolic flux modifications, is Myc-dependent. Myc-driven metabolic reprogramming during tumor progression is modulated by these two proteins [255]. Furthermore, increases in MondoA and ChREBP protein levels lead to Mlx (Max-like protein X) sequestration, promoting the competition between Myc and Mxd (Max dimerization) proteins for Max. Therefore, an imbalance in the Myc protein network may generate metabolic alterations typical of cancer cells. Given the high levels of intermediates generated, Myc also allows shunting from one metabolic pathway to another. Indeed, the glycolytic intermediates glucose-6-phosphate and 3-phosphoglycerate may be used by the pentose phosphate pathway to generate NADPH and ribose for nucleotide biosynthesis [256,257] and serine, which is ultimately converted into NADPH in mitochondria [258].

Myc-enhanced nucleotide biosynthesis sustains the high rate of proliferation characterizing cancer cells. A number of studies have demonstrated Myc-dependent regulation of both purine and pyrimidine [259,260]. Through the pentose phosphate pathway, Myc produces ribose-5-phosphate and induces phosphoribosyl pyrophosphate synthetase 2 (PRPS2) and the synthesis of phosphoribosyl pyrophosphate, serving as scaffold for purine biosynthesis and for the pyrimidine rescue pathway [261]. Myc also supports the induction of genes of metabolic cycles indispensable for nucleotide metabolism, such as the folate cycle and one-carbon metabolism [257,262], orchestrating the control of the expressions of genes participating in different steps of nucleotide synthesis.

The uptake of essential amino acids (EEAs), both large neutral and branched-chain, which serve to build macromolecules and to activate mTOR, requires the expression of families of transporters and enzymes that are under the transcriptional control of Myc [263,264]. Myc is also involved in the alteration of tryptophan metabolism, driving its conversion into kynurenine by upregulating the transporters SLC7A5 and SLC1A5 and the arylformamidase enzyme [253]. Notably, kynurenine was related to enhanced proliferation, migration, and immune escape capacity of tumor cells [265]. Cancer cell lipid and fatty acid metabolism rewires to promote membrane biogenesis and energy storage in fasting, proliferating cells. Myc has been found to be involved in lipid metabolism, promoting both fatty acid–cholesterol synthesis and fatty acid oxidation (FAO). By upregulating enzymes of the TCA cycle, Myc induces the synthesis of citrate, the precursor of fatty acids, as well as a number of enzymes involved in fatty acid synthesis, such as ATP citrate lyase (ACLY), Acetyl-CoA carboxylase (ACACA), fatty acid synthase (FASN), and stearoyl-CoA desaturase (SCD) [266,267,268,269]. Myc also cooperates with regulators of fatty acid synthesis such as MondoA [270] and sterol response element-binding protein 1 (SREBP1) [266]. Furthermore, a positive Myc-dependent feedback loop has been hypothesized to sustain cholesterol metabolism and malignant transformation in cancer cells. Indeed, Myc was demonstrated to induce the expression of 3-hydroxy-3-methyl-glutarylcoenzyme A reductase (HMGCR) [271], which is essential for cholesterol synthesis during tumorigenesis. Conversely, HMGCR phosphorylates and activates Myc in at least some cancer models [272]. Surprisingly, Myc also participates in FAO, typically exploited by healthy cells to produce energy in mitochondria. In this context, Myc activates enzymes required for FAO [273,274], induces the expressions of receptors located both on the cell membrane and on the mitochondrial inner membrane necessary for the uptake of fatty acids undergoing oxidation in mitochondria, and altering Ca^++^ levels [274]. Suppression of FAO in cancer cells occurs through Myc downregulation of a series of enzymes critical for this process [269].

### 5.1. Metabolic Control of Myc Activity in Glioblastoma

Taking into account that the metabolic control of Myc expression in GBM and in other cancer types is largely similar [241,242,243,244,245,246], one of the best recognized and characterized metabolic pathways regulating Myc in GBM is glycolysis, which exerts its modulatory role on Myc expression especially through the mTORC2 component of the mTOR complex [275]. In cancer cells, as stated above, glycolysis and glutaminolysis may be linked through the hexosamine biosynthetic pathway, and glutamine may be converted to glucosamine-6-phosphate and glutamate starting from fructose-6-phosphate and glutamine, a reaction catalyzed by fructose-6-phosphate aminotransferase (GFAT) [276]. It was demonstrated that, in GBM, mTORC2 promotes GFAT1 activity, enhancing glucosamine-6-phosphate synthesis independently of PI3K/Akt signaling. Mechanistically, high glucose and glutamine levels promote mTORC2 activity, which supports Myc protein function as a transcriptional regulator of GFAT1, pushing glucosamine-6-phosphate synthesis [277]. Furthermore, mTORC2 inhibits the phosphorylation of class-IIa HDACs, leading to the acetylation of FOXO1 and 3, the release of Myc from a inhibitory mir34c-dependent pathway, and the activation of aerobic glycolytic genes [278]. Aurora kinase A is another regulator of Myc-dependent glycolytic gene expression in GBM. Indeed, Aurora kinase A inhibition rewires glycolytic metabolism to OXPHOS and FAO through Myc downregulation and the impaired expression of glycolytic genes [279]. Myc is also epigenetically regulated by pyruvate kinase M2 (PKM2) and phosphofructokinase 1 platelet isoform (PFKP), both catalyzing pivotal steps in the glycolytic cycle. In EGFR-mutated GBM, PKM2 translocates to the nucleus, causing the dissociation of HDAC3 from the Myc promoter through the phosphorylation of Threo11 on histone H3, inducing its transcription [280]. PFKP induces β–catenin translocation via EGFR-signaling activation and transcriptional enhancement of the downstream effectors CCND1 and Myc [281]. Both processes further promote glycolysis, glutaminolysis, and cell proliferation. Factors controlling FAO, such as peroxisome-proliferator-activated receptor α (PPRα), have also been reported to regulate Myc expression in GBM. Indeed, PPRα depletion led to a downregulation in Myc expression in GSCs [282].

### 5.2. Myc-Dependent Regulation of Metabolic Pathways in Glioblastoma

As in other tumors, in this malignant brain cancer Myc activity underlies the control of glycolysis and glutaminolysis. Recent studies have revealed novel partners and effectors of Myc in determining metabolic rewiring in GBM. A role for importin-α1 (also known as karyopherin α2, KNAP2) in the regulation of Myc activity and glycolysis has been found. Indeed, importin-α1 leads to E2F1 nuclear translocation and Myc transcription activation in GBM cells. This, in turn, promotes the expression of Myc-regulated enzymes involved in glucose metabolism, such as Glut1, HK2, PKM2, and PFK1 [283]. As stated above, Myc induces the expressions of the glycolytic genes *Glut1*, *Glut3*, *PDK1*, and *HK2* in EGFRvIII^+^GBM in a self-sustaining loop based on the Myc-dependent upregulation of splicing factor hnRNPA1. The latter provides the production of a spliced isoform of Max called Delta-Max, which depends on the availability of glucose. By forming functional heterodimers with Myc, Delta-Max binds to the *Glut1*, *Glut3*, *PDK1*, and *HK2* promoters, enhancing the expressions of respective enzymes. Intriguingly, while Delta-Max is able to rescue *Glut1*, *Glut3*, *PDK1*, and *HK2* expressions upon EGFRVIII depletion, wild-type Max is not, and consistently does not support EGFRvIII^−^ GBM cell proliferation in the presence of glucose. In parallel, Delta-Max knockout inhibits the expressions of glycolytic genes and reduces the size of GBM tumors in vivo [133]. By cooperating with the aryl hydrocarbon receptor (AHR), a detoxifying cytoplasmic receptor [284,285], Myc regulates glycolysis and pyrimidine biosynthesis in GBM cells. In fact, AHR knockdown altered the levels of 26 metabolites belonging to redox equilibrium and fatty acid and nucleotide metabolism. Binding of the Myc/Max and AHR/aryl hydrocarbon receptor nuclear translocator (ARNT) heterodimers on the promoter regions of CAD, DHODH, UMPS, and LDHA has been postulated [286], and in vivo lactate labeling overlaps Myc expression in patient-derived GBM samples [287]. At the epigenetic level, Myc binds to superenhancers in GBM glycolytic genes. The disruption of superenhancers by HDAC inhibitors (HDACis) impairs glycolysis, with a shift toward OXPHOS and FAO driven by the PGC1α and PPARD genes and paralleled by a decrease in Myc activity (due to transcriptional repression), which in normal conditions blunts HDAC2, PGC1α, and PPARD expression. These metabolic alterations lead to a decrease in ATP production and a compensatory increase in oxygen consumption rate (OCR). U-13C-glucose carbon tracing shows that not only glycolytic intermediates and other associated metabolites are low in HDACi-treated GBM cells, but also PPP, ribose, and serine production, as well as the hexosamine pathway and lipid synthesis, all of them related to the Warburg effect [288].

Myc also controls the expressions of the kidney-type isoforms of glutaminase KGA and CAG encoded by the *GLS* gene associated with cell proliferation, whereas the liver-type isoforms GAB and LGA encoded by the *GLS2* gene and related to quiescence and are not detectable in GBM. Indeed, forcing GLS2 expression in GBM cells slowed their proliferation and potentiated the antiproliferative effect of GLS silencing [289]. In other tumor types, Myc upregulates mitochondrial glutaminase through the transcriptional repression of mir23a/b and the consequent upregulation of its target, mitochondrial glutaminase [248]. Although the mechanisms have not been yet established in glioblastoma, neurons express mir23a/b, and mir23a controls EMT in GBM by targeting homeobox D10 (HOXD10) [290], whereas mir23b is downregulated by Myc in ischemic neurons, leading to nuclear factor erythroid 2-related factor 2 (Nrf2) upregulation and reductions in apoptosis and the infarcted area [291]. Furthermore, Myc has been found located on the mir23b promoter by chromatin immunoprecipitation, suggesting a direct Myc-dependent transcriptional mechanism [292]. Therefore, we may argue that similar transcriptional pathways could occur in GBM, leading to mitochondrial glutaminase upregulation.

The Myc-dependent synthesis of different metabolic products has a profound impact on the maintenance of GSCs. Indeed, Myc has been found to regulate mevalonate signaling in GSCs. Coenzyme-Q and cholesterol are the end products of the mevalonate pathway, together with isoprenoid intermediates, serving as signal transducers [293]. GSCs typically overexpress mevalonate pathway enzymes and silencing of the rate-limiting enzyme HMGCR, which converts HMG-CoA to mevalonic acid, dramatically impairs GSCs proliferation ability and viability. Affecting protein prenylation with farnesyl and geranylgeranyl transferase inhibitors blocks GSC viability and neurosphere formation. Myc has been found recruited at the promoters of the six major enzymes of the mevalonate pathway, and its depletion leads to decreases in their expressions that are paralleled with reduced GSC viability, self-renewal, and proliferation, indicating the Myc-promoted mevalonate pathway as pivotal for GSC maintenance [294]. Purine synthesis, which is strictly connected to glucose availability and uptake through the GLUT3 transporter [127], is another metabolic pathway controlled by Myc in GSCs. Myc KO in GSCs results in decreased expressions of purine biosynthetic enzymes and the IMP, AMP, and GMP metabolites, whilst in GSCs committed to differentiation, high levels of Myc enhance the expressions of purine biosynthetic enzymes and related metabolites. Mechanistically, Myc has been found located on promoters of *IMPDH1*, *PPAT*, *PRPS1*, *ADSS*, *ADSL*, and *GMPS* and interfering with the expressions of enzymes devoted to purine biosynthesis, strongly limiting GSC self-renewal ability [295]. Glucose is not the only source of carbon in cancer metabolism. Indeed, glutamine partially rescues GSC viability upon glucose deprivation and serves also as a source of nitrogen atoms for purine synthesis. By controlling glutaminolysis, Myc provides both intermediates and nitrogen atoms to fulfill GSC demands for the TCA cycle and purine synthesis, respectively.

#### Crosstalk between Myc-Dependent Angiogenesis and Metabolism in GBM

GBM is one of the most vascularized tumors, and GBM cells exploit different ways to ensure the adequate uptake of nutrients and oxygen from the microenvironment. In addition to the well-known process of neoangiogenesis, the formation of new blood vessels from pre-existing ones, GBM cells deploy vascular mimicry and vascular co-options to both increase blood supply and invade surrounding tissue (for details, see Rosiňska and Gavard, 2021 [296]). GBM is characterized by extremely hypoxic and necrotic areas. The hypoxic environment characterizing a tumor mass, with the activation of HIF1α and the downstream transcriptional cascade leading to the production of a plethora of pro-angiogenic factors, is one of the major determinants of GBM neoangiogenesis [297]. Once activated, this pathway not only promotes the remodeling of the vasculature through extracellular matrix degradation via metalloproteinases and the activation of stromal cells, as well as the recruitment, migration, and proliferation of endothelial cells [298], but it also induces GSCs to differentiate into endothelial-like cells, forming capillary-like structures [299]. The latter have a larger lumen with respect to their normal counterparts and present irregular branching [300]. Leakiness frequently occurs, causing plasma extravasation and focal edema [301]. At the molecular level, in addition to the well-established mechanism relying on the O_2_-deprivation-dependent induction of HIF1α [302], it has been recently established that, upstream of HIF1α, lncRNA H19 regulates glioma angiogenesis. Acting as a sponge for mir138, which targets HIF1α, lncRNA H19 promotes glioma angiogenesis in a HIF1α- and vascular endothelial growth factor (VEGF)-dependent manner [303]. Other noncoding RNAs control GBM angiogenesis [304], as well as transcription factors such as NK-kB [305] and STAT3 [306].

In this scenario, Myc participates in GBM angiogenesis regulation, even in normoxia. In this condition, wnt/β-catenin pathway activation leads to Myc overexpression and consequent HIF1α upregulation independently of O_2_ deprivation. β-catenin/TCF4 in complex with STAT3 also activates HIF1α, which in turn upregulates Myc in a positive feedback loop [307].

EGFRvIII^+^ GBM has been found to be more vascularized than EGFRwt tumors in mouse xenograft models. In this condition, Myc also plays a pivotal role in driving tumor angiogenesis. Indeed, the EGFRvIII-dependent upregulation of Myc promotes ANG4 transcription through direct binding to its promoter [214]. Myc also blunts the production of antiangiogenic factors in GBM, thus promoting angiogenesis indirectly. In fact, one of the most well-known Myc target loci, the mir17-92 cluster [308], was reported to downregulate the expressions of transforming growth factor receptor (TGFRβ) II and Smad4, impairing the production of antiangiogenic molecules such as thrombospondin and clusterin [309].

Very recently, Myc was also found to induce GBM angiogenesis through histamine production and the release of this metabolite in the tumor microenvironment from GSCs. Indeed, Myc binds the histidine decarboxylase (HDC) promoter, which in GSCs is in an open chromatin conformation, bearing a high amount of trimethylated histone H4 on lysine 4 (H3K4me3), as revealed by ATACseq experiments. Released histamine in the extracellular space binds to the histamine H1 receptor on endothelial cells, activating the NF-kB pathway and angiogenesis [310]. Furthermore, the activation of Myc/HIF1α-dependent aerobic glycolysis gene expression pushes the production of lactate, which is released by GBM cells in the extracellular space, inducing acidosis. The latter activates HIF1α in the surrounding endothelial cells with the production of proangiogenic molecules, such as VEGF [307]. These results indicate a strict interrelationship among gene expression, metabolism, and final biological outcomes (i.e., angiogenesis), with Myc as one of the most important molecular coordinators.

### 5.3. Targeting Myc-Dependent Metabolic Pathways in GBM

Targeting cancer metabolism for therapeutic purposes is a hot topic. Glucose metabolism is the pathway toward which a large effort has been made, and many antiglycolytic molecules have been investigated in preclinical studies or have entered phase-I and -II clinical trials [311]. These agents target glycolytic enzymes, such as HK2 [312,313], GAPDH [314,315], PDK [316,317,318], and LDHA [319,320], or glucose transporters [321,322,323]. Interestingly, most of them are under the transcriptional control of Myc, as well as GLS1, which has become a putative therapeutic target for cancer in the past few decades [324], with at least one inhibitor (CB-839 [325,326]) used in 21 clinical trials on a variety of tumors (including ovarian cancer, renal carcinoma, breast cancer, colorectal cancer, non-small-cell lung cancer, and IDH-mut grade-II and -III astrocytomas; https://www.clinicaltrials.gov/ct2/results?cond=Cancer&term=CB-839&cntry=&state=&city=&dist=&Search=Search; accessed on 12 January 2023), mainly in combination with other chemotherapeutic agents. The feasibility of targeting glycolysis in malignant brain tumors was demonstrated in the early 1990s [312] and validated in the past few years [316]. Given the dependency of GBM on Myc-driven enhanced glycolytic flux, recently Myc-deregulated glycolysis has been evaluated as a putative therapeutic target in preclinical models of GBM. Specifically, it has been demonstrated that forcing Myc expression in GBM cells characterized by low levels of Myc protein strongly enhances the production of glycolytic intermediates and induces glucose dependence for cell growth. Consistently, in patient-derived GBM cells typically expressing high levels of Myc, Myc KO led to downregulations in glycolytic enzymes and related products [327]. On these bases, small molecule inhibitors of nicotinamide phosphoribosyl-transferase (NAMPT) essential for the NAD^+^ requiring glycolytic GAPDH step have been employed both in vitro and in vivo as a proof-of-concept for Myc-driven glycolytic targeting in GBM. Specifically, FK866 [328] and GMX1778 [329] each induced apoptotic cytotoxicity in GBM cells in a Myc-dependent manner by glycolysis inhibition, and orally administered GMX1778 significantly extended the survival of mice undergoing Myc-amplified patient-derived orthotopic xenografts [327]. These findings indicate Myc-controlled metabolic routes as valuable and significant therapeutic targets to take into account for GBM treatment.

## 6. Conclusions: Postulating a Myc-Centered Metabolic Circuit in Glioblastoma

The information provided within this work clearly demonstrates how Myc is central for almost all aspects of cell biology, either healthy or cancerous, as outlined below.

I.Coordinating a plethora of transcriptional coregulators, especially RNAP I, II, and III, as well as controlling RNA splicing and capping, Myc activity is not limited to the activation or repression of directly bound genes at promoters. Furthermore, by binding to superenhancers and increasing chromatin contacts at transcriptionally active domains, Myc also governs chromatin topology [330]. Taking into account all this information, Myc may be defined as a “superoncogene”, orchestrating chromatin architecture, modulating RNAP activity, and directly inducing or repressing transcription, thus ruling the expression of gene expression programs and, ultimately, determining cell fate.II.Myc involvement in metabolic processes further underpins its importance in the maintenance of cellular homeostasis and in the metabolic rewiring of cancer cells where it is largely overexpressed. In GBM, multiple metabolic signals control Myc expression, which, in turn, governs the activation of cancer metabolic routes and the shunting from one pathway to another (Figure 3), suggesting a Myc-centered, self-sustaining metabolic circuit fulfilling GBM cell and, more importantly, GSC energy demand, fostering tumor growth.III.The launching and completion of many clinical trials targeting cancer cell metabolism in a variety of cancer types using small molecules that hit pivotal Myc-dependent metabolic enzymes and pathways, including astrocytomas and gliomas, highlights the importance of providing an in-depth characterization of this Myc-centered circuit in GBM as a way to design novel therapeutic strategies aimed at increasing the pool of weapons against this deadly type of tumor.

## Figures and Tables

**Figure 1 ijms-24-04217-f001:**
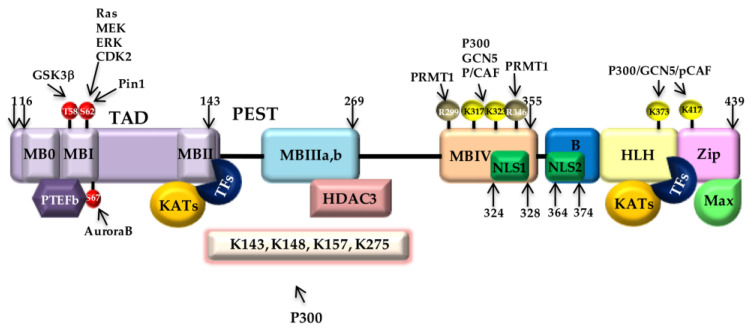
The complex structure of the Myc oncoprotein. Structures of distinct Myc domains, relative post-translationally modified residues, and interacting regions with protein partners.

**Figure 2 ijms-24-04217-f002:**
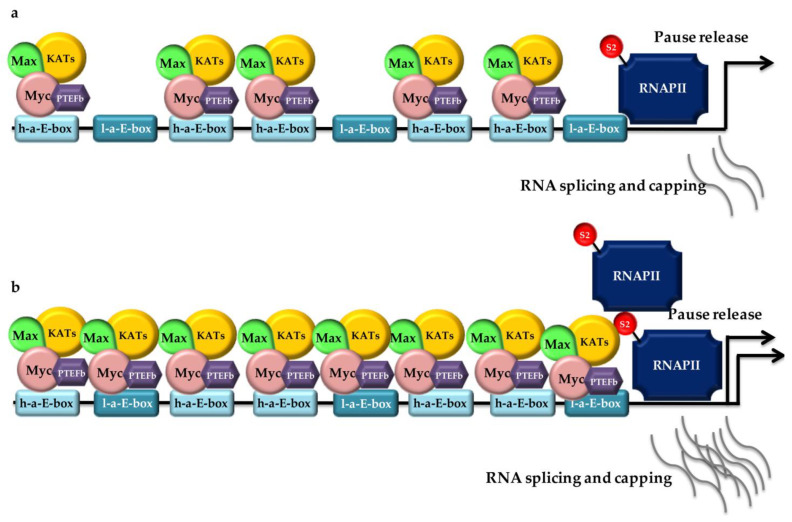
Myc regulation of gene transcription in healthy and cancer cells. Schematic representation of abundancy-dependent Myc binding at target promoters. (**a**) In healthy cells, Myc/Max and coregulators preferentially bind to high-affinity E-boxes (h-a-E-box). By inducing both the opening of chromatin through histone acetylation and RNAPII pause and release via PTEFb-dependent phosphorylation of Ser2 on RNAPII, Myc/Max heterodimers tune the transcription, splicing, and RNA capping of genes pivotal to cellular homeostasis. (**b**) In overexpressing Myc cancer cells, Myc/Max heterodimers also invade low-affinity binding sites (l-a-E-box) and push transcription, leading to the execution of aberrant gene expression programs.

**Figure 3 ijms-24-04217-f003:**
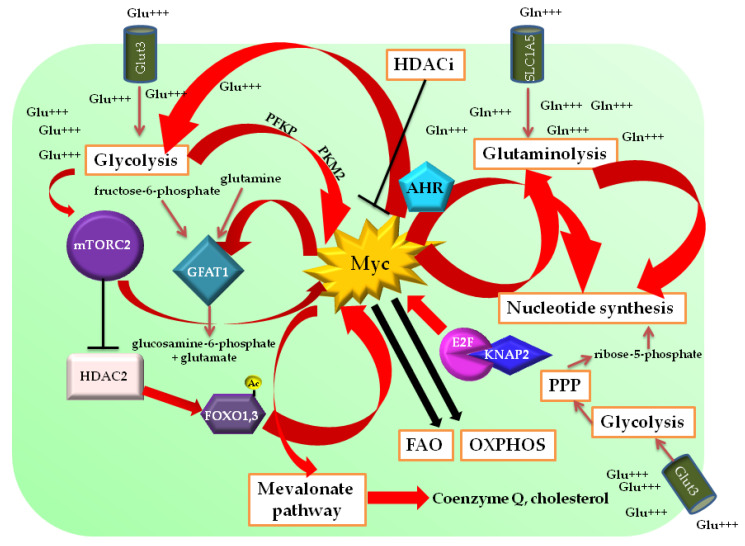
Myc at the intersection between metabolism and gene expression in glioblastoma. The image depicts the major metabolic pathways that regulate and are regulated by Myc in GBM. Basically, Myc controls all the metabolic pathways in GBM, whereas glycolysis remains the main controller of Myc activity in this type of brain tumor. Glu—glucose; Gln—glutamine.

**Table 1 ijms-24-04217-t001:** Clinical trials related to Myc overexpressing brain tumours.

Trial N°	Status	Study Title	Drug/Treatment	Condition	Sponsor	Collaborators
NCT03434262	Active, not recruiting	SJDAWN: St. Jude’s Children Research Hospital Phase-I study evaluating molecularly driven doublet therapies for children and young adults with recurrent brain tumors	Ribociclib/trametinib, ribociclib/sonidegib, ribociclib/gemcitabine	Brain tumors, including medulloblstoma, ependymoma, oligodendroglioma, glioblastoma (IDHwt and IDHmut), xanthoastrocytoma, neuroblastoma, medulloepithelioma, and embryonal tumors	St. Jude’s Children Research Hospital	Novartis Pharmaceuticals
NCT03224104	Completed	Study of TG02 in elderly newly diagnosed or adult relapsed patients with anaplastic astrocytoma or glioblastoma	TG02, TMZ, radiation	IDH1R132H-non mutated and MGMT-promoter-unmethylated anaplastic astrocytoma or GBM; IDH1R132H-non mutated and MGMT-promoter-methylated anaplastic astrocytoma or GBM	European Organization for Research and Treatment of Cancer (EORTC)	Tragara Pharmaceuticals
NCT02711137	Terminated	Open-label safety and tolerability study of INCB057643 in subjects with advanced malignancies	INCB057643, gemcitabine, paclitaxel, rucaparib, abiraterone, azacitidine, ruxolitinib	Solid tumors, including brain tumors and lymphoma	Incyte Corporation	Incyte Corporation

## Data Availability

Not applicable.

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
