# Peer review of "The “Superoncogene” Myc at the Crossroad between Metabolism and Gene Expression in Glioblastoma Multiforme"

_ijms, 2023, doi:10.3390/ijms24044217_

Round 1
Reviewer 1 Report
The review article by Cencioni et al. (The “superoncogene” Myc at the crossroad between metabolism and gene expression in glioblastoma multiforme) is an interesting paper in whichi the authors put a large effort.
The role of c-myc in glioblastoma metabolism has been in deep analyzed. I have some suggestions to improve the scientific quality of the paper.
1.The impact on therapy has not discussed in detail. Can the authors enlist, in a Table, agents of possible therapeutic interest able to interfere with some of the described Myc-related metabolic pathways? This could be of interest for the reader. See for instance: Teteishi et al., Myc-driven glycolysis is a therapeutic target in glioblastoma, 2017. Are any clinical trials ongoing?
2. Can the authors comment about the efficacy of myc-targeting in clinical trials? Despite the key role of Myc (the “Superoncogene”) few clinical trials are ongoing on glioblastoma patients. Please comment and include a section on clinical trials targeting myc in GBM, if they are available.
3. In my opinion, considering the large amount of information andd how it is presented within the review paper, a short “Conclusion section” perhaps divided in “bullet points” is needed.
Thank you for the opportunity to review this interesting study.
Author Response
The authors wish to thank the Reviewer for the positive evaluation of their review MS and helpful suggestions. Below are our point-to-point answers to Reviewers’ comments. Changes in the MS are tracked in red colour.
Q1.The impact on therapy has not discussed in detail. Can the authors enlist, in a Table, agents of possible therapeutic interest able to interfere with some of the described Myc-related metabolic pathways? This could be of interest for the reader. See for instance: Teteishi et al., Myc-driven glycolysis is a therapeutic target in glioblastoma, 2017. Are any clinical trials ongoing?
A1. We thank the Reviewer for this observation. To fulfill this Reviewer request we have added the paragraph 5.3 in the revised version of the MS, where we discuss the targeting of Myc-driven cancer metabolism for therapeutic purposes. A table of agents of therapeutic interest acting on different cancer metabolic routes, indicating also whether and which of them are employed in clinical trials, has been very recently published (see ref. 290 of the new MS). Therefore, we have preferred to enlist in a table the clinical trials specifically targeting Myc in GBM, now included in the new subparagraph 4.4 of the revised version of the MS.
Q2. Can the authors comment about the efficacy of myc-targeting in clinical trials? Despite the key role of Myc (the “Superoncogene”) few clinical trials are ongoing on glioblastoma patients. Please comment and include a section on clinical trials targeting myc in GBM, if they are available.
A2. As mentioned above, a specific subparagraph (4.4) and a table discussing this issue has been added at page 16 of the new MS.
Q3. In my opinion, considering the large amount of information and how it is presented within the review paper, a short “Conclusion section” perhaps divided in “bullet points” is needed.
A3. We thank the Reviewer for this comment. Indeed, a Conclusion section (n° 6) was present also in the previous version of the MS. Now we have divided this section in bullet points as requested.
Reviewer’s final comment: Thank you for the opportunity to review this interesting study.
Authors response: we are extremely grateful to this Reviewer for having appreciating our effort aimed to put under the spotlight and possibly link many aspects of GBM biology which are under the control of the Myc oncogene.

Reviewer 2 Report
Your manuscript entitled "The “superoncogene” Myc at the crossroad between metabolism and gene expression in glioblastoma multiform" has been reviewed. Authors summarized the available information on GBM metabolism with a specific focus on the control of Myc oncogene. In addition, the activation of metabolic signals, ensuring GBM growth.
The article submitted for review is an interesting contribution to further research, and significant efforts have been made in this work. Therefore, the findings presented in this review may be useful for the scientific community. In this study, data were collected from 265 international references about Myc oncogene, transcription factors, Glioblastoma, and metabolic pathways.
To improve the manuscript there are some suggestions as follow:
1. In line, 211 add reference [9].
2. In lines 133,134 the references should be changed to “in Verhaak, R.G., Cell, 2010 and TCGA network, Nature 2008[5,40]–“
3. In line 378 remove the extra bracket
4. Insufficient information about the relationship of Myc to Apoptosis. Therefore, it is important to collect and add information about the processes, pathways and molecular factors that are involved in apoptosis, especially those associated with Myc such as the Bcl-2, P53, Caspases family, and other factors.
Author Response
We wish to thank the Reviewer for the highly positive evaluation of our MS and very helpful suggestions. Below are our point-to-point answers to Reviewer’s comments. Changes in the MS are tracked in red colour.
Q1. In line, 211 add reference [9].
A1. We thank the Reviewer for this comment. Reference 9 has been added to line 211.
Q2. In lines 133,134 the references should be changed to “in Verhaak, R.G., Cell, 2010and TCGA network, Nature 2008[5,40]“
A2. As requested, we have changed references at lines 133 and 134.
Q3. In line 378 remove the extra bracket.
A3. We thank the Reviewer for this careful observation. The extra bracket has been removed.
Q4. Insufficient information about the relationship of Myc to Apoptosis. Therefore, it is important to collect and add information about the processes, pathways and molecular factors that are involved in apoptosis, especially those associated with Myc such as the Bcl-2, P53, Caspases family, and other factors.
A4. We thank the Reviewer for raising this issue. A specific discussion on Myc-dependent regulation of apoptosis has been included in the new sub-paragraph 4.1.5 at page 13 of the new MS.

Reviewer 3 Report
The manuscript reviews the roles of Myc in promoting tumor growth in glioblastoma, as well as the metabolic adaptation that takes place in GBM to enable survival, further growth and progression. GBM remains a challenging cancer to treat and a better understanding of the mechanisms of disease progression and how molecules such as Myc may affect response to treatment are relevant.
The authors tend to use slogan-like phrases (exciting great body of knowledge, master regulator and great target), which make parts of an otherwise well-written paper to read like an advert/commercial. The authors should rather keep the writing scientific.
Heading 3.3 should be lipid metabolism, because fa's are lipids.
The authors should elaborate on mechanisms when addressing metabolic pathways. This is the case under lipid metabolism where it is not clear which key enzymes are regulated by Myc and how. The same is observed with glycolysis where contribution to the regulation of key control points is missing. For example, the authors state that alternative splicing of Myc-associated X results in Delta max formation, which in turn supports glycolytic gene expression. This is a high level overview that does not unpack mechanism, and it should.
The authors combine growth and metabolism in GBM in the same paper, which makes the paper interesting as these are in essence very intertwined. What would make the manuscript compelling is addressing the aspect of angiogenesis. GBM is a highly vascular tumor, and the abnormal vasculature contributes to the challenge of treatment. Moreover, there is an intricate relationship between angiogenesis, metabolic adaptation and neoplastic growth.
The manuscript requires editing, there are a few grammatical mistakes and typing errors; the authors should also decide between UK English and USA English.
Author Response
IJMS-216054 Authors’ response to Reviewer 3
The authors wish to thank this Reviewer for helpful criticisms and suggestions.
Below authors point-to point answers to Reviewer’s comments are reported. Changes to the MS are tracked in red colour.
Q1. The authors tend to use slogan-like phrases (exciting great body of knowledge, master regulator and great target), which make parts of an otherwise well-written paper to read like an advert/commercial. The authors should rather keep the writing scientific.
A1. We thank the Reviewer for this comment. We have modified the text accordingly.
Q2. Heading 3.3 should be lipid metabolism, because fa's are lipids.
A2. We thank the Reviewer for this observation. We have changed the title of subparagraph 3.3.
Q3. The authors should elaborate on mechanisms when addressing metabolic pathways. This is the case under lipid metabolism where it is not clear which key enzymes are regulated by Myc and how. The same is observed with glycolysis where contribution to the regulation of key control points is missing. For example, the authors state that alternative splicing of Myc-associated X results in Delta max formation, which in turn supports glycolytic gene expression. This is a high level overview that does not unpack mechanism, and it should.
A2. We thank the Reviewer for this comment. Indeed, molecular mechanisms of Myc-dependent control of metabolic gene expression are reported in subparagraph 5.2, page 19, (a. control of Myc by importin1a and E2F with consequent transcriptional upregulation of glycolytic genes; b. cooperation of Myc with AHR; c. Myc-dependent epigenetic control of glycolytic genes’ super-enhancers); page 20, (recruitment of Myc to mevalonate pathway gene promoters). We have further addressed in this section the molecular mechanism underlying Delta-Max production and glycolytic genes upregulation(page 19,) Myc-dependent control of glutaminolysis (page 20) and purine biosynthesis (page 21). Enzymes involved in lipids biosynthesis and whose expression is regulated by Myc have been added at page 18. We have focused on molecular mechanisms specifically occurring in GBM. Mechanisms related to Myc-dependent control of cancer metabolism, including lipid metabolism, have been extensively reported elsewhere (Stine ZE et al.,, Cancer Discovery, 2015;Tambay V. et al., Cancers, 2021; Dong Y. et al., Signal Transduct Target Ther. 2020; Hsieh AL et al., Semin Cell Dev Biol. 2015).
Q4. The authors combine growth and metabolism in GBM in the same paper, which makes the paper interesting as these are in essence very intertwined. What would make the manuscript compelling is addressing the aspect of angiogenesis. GBM is a highly vascular tumor, and the abnormal vasculature contributes to the challenge of treatment. Moreover, there is an intricate relationship between angiogenesis, metabolic adaptation and neoplastic growth.
A4. We really thank the Reviewer for this comment, which further highlights Myc as regulator of basically all the aspects of cancer biology. We have added an additional subparagraph (5.2.1) to provide a brief overview of Myc –dependent regulation of GBM angiogenesis and its crosstalk with metabolism, which is, in our opinion, a topic deserving a separate review article.
Q5. The manuscript requires editing, there are a few grammatical mistakes and typing errors; the authors should also decide between UK English and USA English.
A5. We have further edited the MS.

Round 2
Reviewer 3 Report
The authors have addressed the recommendations made. I support the publication of the manuscript in its current state.